# Preferent Diaphragmatic Involvement in TK2 Deficiency: An Autopsy Case Study

**DOI:** 10.3390/ijms22115598

**Published:** 2021-05-25

**Authors:** Sara Laine-Menéndez, Cristina Domínguez-González, Alberto Blázquez, Aitor Delmiro, Inés García-Consuegra, Miguel Fernández-de la Torre, Aurelio Hernández-Laín, Javier Sayas, Miguel Ángel Martín, María Morán

**Affiliations:** 1Mitochondrial and Neuromuscular Diseases Laboratory, Instituto de Investigación Sanitaria Hospital ‘12 de Octubre’ (‘imas12’), 28041 Madrid, Spain; slaine.imas12@h12o.es (S.L.-M.); cdgonzalez@salud.madrid.org (C.D.-G.); abencinar@hotmail.com (A.B.); adelmiro@h12o.es (A.D.); inesgcg@hotmail.com (I.G.-C.); Miguel.fnandezt.imas12@h12o.es (M.F.-d.l.T.); mamcasanueva.imas12@h12o.es (M.Á.M.); 2Spanish Network for Biomedical Research in Rare Diseases (CIBERER), U723, 28029 Madrid, Spain; 3Deparment of Neurology, Neuromuscular Unit, Hospital 12 de Octubre, 28041 Madrid, Spain; 4Department of Pathology (Neuropathology), Hospital 12 de Octubre, 28041 Madrid, Spain; aurelio.hlain@salud.madrid.org; 5Department of Pneumology, Ventilation Unit, Hospital 12 de Octubre, 28041 Madrid, Spain; jsayascat@gmail.com

**Keywords:** mitochondrial diseases, diaphragm, thymidine kinase 2, respiratory failure

## Abstract

Our goal was to analyze *post*
*mortem* tissues of an adult patient with late-onset thymidine kinase 2 (TK2) deficiency who died of respiratory failure. Compared with control tissues, we found a low mtDNA content in the patient’s skeletal muscle, liver, kidney, small intestine, and particularly in the diaphragm, whereas heart and brain tissue showed normal mtDNA levels. mtDNA deletions were present in skeletal muscle and diaphragm. All tissues showed a low content of OXPHOS subunits, and this was especially evident in diaphragm, which also exhibited an abnormal protein profile, expression of non-muscular β-actin and loss of GAPDH and α-actin. MALDI-TOF/TOF mass spectrometry analysis demonstrated the loss of the enzyme fructose-bisphosphate aldolase, and enrichment for serum albumin in the patient’s diaphragm tissue. The TK2-deficient patient’s diaphragm showed a more profound loss of OXPHOS proteins, with lower levels of catalase, peroxiredoxin 6, cytosolic superoxide dismutase, p62 and the catalytic subunits of proteasome than diaphragms of ventilated controls. Strong overexpression of TK1 was observed in all tissues of the patient with diaphragm showing the highest levels. TK2 deficiency induces a more profound dysfunction of the diaphragm than of other tissues, which manifests as loss of OXPHOS and glycolytic proteins, sarcomeric components, antioxidants and overactivation of the TK1 salvage pathway that is not attributed to mechanical ventilation.

## 1. Introduction

Recessive mutations in the thymidine kinase 2 (*TK2*) gene (MIM# 188250) cause mitochondrial DNA (mtDNA) depletion and multiple mtDNA deletions [1]. TK2 is a mitochondrial matrix deoxyribonucleoside kinase involved in mtDNA precursor synthesis by catalyzing the phosphorylation of pyrimidine nucleosides. Mitochondrial deoxynucleoside triphosphate (dNTP) pools required for DNA synthesis are produced by *de novo* or nucleotide salvage pathways, which are differentially utilized in proliferative and post-mitotic cells. TK2 is essential in the mitochondrial salvage pathway in postmitotic (quiescent) cells, which downregulate the cytosolic *de novo* cytosolic pathway especially in skeletal muscle [2]. In proliferating cells, the salvage pathway is based on cytosolic thymidine kinase 1 (TK1) and deoxycytidine kinase (dCK), which phosphorylate deoxythymidine and deoxycytidine, respectively.

TK2 deficiency manifests predominantly as a progressive myopathy with a broad spectrum of severity ranging from extremely severe and rapidly progressive early-onset forms (with a survival of less than two years) to less severe forms with a childhood, late or very late-onset, and a slower rate of progression, but with an almost invariable respiratory involvement, which appears during disease progression, and shortens life expectancy [3,4,5]. The mildest form of the clinical spectrum described so far is a myopathy that manifests only as recurrent rhabdomyolysis and exercise intolerance, for at least the first 10 years of evolution, of which only one case has been described in the literature [6]. Late-onset TK2 deficiency is the least common form, and its natural history is not well understood. We previously suggested that there is early, selective and severe involvement of the diaphragm in most patients with late-onset TK2 deficiency, based on clinical symptoms and detailed respiratory assessment [5,7], which has also been demonstrated by direct functional analysis of diaphragmatic function [8].

To date, only two early-onset cases with severe TK2 deficiency have been autopsied. The first study reported histological brain alterations in four patients including oedema, neuronal loss, gliosis and ischaemic injury that were associated to mtDNA depletion only in two of the children analyzed [9]. A second study described severe myopathy and mtDNA depletion in skeletal muscle, brain demyelination, liver steatosis and hypertrophic cardiomyopathy in a 22-month-old girl harbouring two missense mutations in the TK2 gene [10].

Recently, a treatment based on oral nucleosides (deoxythymidine and deoxycytidine) has proven effective and safe for TK2 deficiency in both preclinical and preliminary clinical evaluations [7,11,12]. Studies in *TK2* knock-out and mutant knock-in mouse models have shown that the effect of deoxypyrimidine supplementation relies on the recruitment of alternative cytosolic salvage pathways for dNTP synthesis: TK1 and dCK [13,14]. Furthermore, it has previously been demonstrated that the postnatal downregulation of *TK1* in the *TK2* mutant knock-in mouse model unmasks TK2 deficiency, and correlates with the onset of mtDNA depletion in tissues [15]. This metabolic change also accounts, at least in part, for the loss of the effect of nucleoside supplementation in adult mice, and may also explain a lower response to treatment in adult patients [13,14].

Here we report, for the first time to our knowledge, the autopsy results of an adult patient with late-onset TK2 deficiency who died of respiratory failure at the age of 43. To gain more insight into the respiratory involvement of the disease, particularly in the diaphragm, we comprehensively analyzed mtDNA copy number, oxidative phosphorylation (OXPHOS) subunit composition, glycolytic enzyme levels, myofibrillar proteins, antioxidant enzymes and proteins involved in protein degradation pathways (autophagy, proteasome and proteases) in several *post mortem* tissues including diaphragm and skeletal muscle. We also analyzed TK2 and TK1 levels in muscle tissue.

## 2. Results

### 2.1. Clinical Summary

The patient was a 43-year-old woman with a mitochondrial myopathy secondary to the homozygous c.323C>T, p.(Thr108Met) mutation in *TK2* (NM_004614.5). The Thr108Met mutation was described in 2002 associated with mtDNA depletion in a patient with myopathy and spinal muscle atrophy-like symptoms and since then, it has been described in several patients having mtDNA maintenance defects [3,16]. Her parents were consanguineous, and a sister died of respiratory failure at 14 years of age during a pneumonia episode some years before. Her sister had had difficulties running, jumping and climbing stairs since childhood, but was never was evaluated by a neurologist or examined in an etiological study. The patient did not show any symptoms until the second decade of life when she started to notice progressive muscle weakness. The first consultation with a neurologist was at age 23. On examination, she had a bilateral facial weakness with no ptosis, and oculomotor movements were normal. She had a nasal voice and mild dysphagia. She was unable to lift her head from the supine position, had mild proximal arm and leg weakness and walked with a waddle. At age 29, she complained of nocturnal hypoventilation symptoms (orthopnea, excessive daytime sleepiness, morning headache, drowsiness and unrestful sleep) and a restrictive respiratory insufficiency was diagnosed, leading to the start of nocturnal non-invasive ventilation (NIV). At age 40, she required a percutaneous gastrostomy due to worsening dysphagia symptoms and very low body mass index. At that time, she used the NIV most of the day but still retained the ability to walk without assistance. A genetic diagnosis was established when she was 42. The patient died of respiratory failure at the age of 43. Before dying, she expressed her desire to donate her organs for research.

Autopsy results: A muscle biopsy obtained from the quadriceps showed numerous ragged-red fibers that were hyper-reactive for succinate dehydrogenase, and most were cytochrome-c oxidase (COX)-negative (COX-negative fibers accounted for ~20% of all fibers). Frequent necrotic fibers, some with myophagocytosis, and scattered regenerative fibers were also identified, and dystrophic features with mild endomysial fibrosis and adipose tissue replacement were also present (Appendix A). Marked type I fiber predominance was also observed. Neuropathological autopsy study included frontal, temporal and occipital cortex and underlying white matter; hippocampus; amygdala; basal ganglia; thalamus and hypothalamus; midbrain; pons and cerebellum; medulla, spinal cord; dorsal root ganglia and dorsal roots and ventral roots. The neuropathological study showed no significant abnormalities. The diaphragm muscle was not morphologically evaluated.

### 2.2. mtDNA and OXPHOS Proteins in TK2 Deficiency

We first analyzed mtDNA content in tissues by real-time PCR. We found that the mtDNA/nDNA copy-number ratio in skeletal muscle, liver, kidney, small intestine, and particularly in diaphragm was lower in TK2-deficient tissues than in control tissues (20%, 24%, 35%, 11% and 4%, respectively), whereas no changes were detected in heart and brain tissue (91% and 88%, respectively) (Figure 1A). Analysis of mtDNA integrity using LR-PCR revealed mtDNA deletions only in the skeletal muscle and diaphragm of the TK2-deficient patient (in apparently low and high proportions, respectively) and not in the liver, kidney and small intestine. As expected, no mtDNA deletions were detected in *post mortem* diaphragm tissue from the ventilated and non-ventilated control patients (Figure 1B).

To examine the impact of the lower levels and deletions of mtDNA on the OXPHOS system, we used Western blotting to assess the steady-state levels of several subunits encoded by nuclear (ATP synthase subunit α, ATP5a; succinate dehydrogenase iron-sulphur subunit, SDHB; and NADH dehydrogenase 1 beta subcomplex 8, NDUFB8) and mitochondrial (cytochrome b-c complex subunit 2, UQCRC2, and cytochrome c oxidase subunit 2, COXII) DNA (Figure 2A). We found that levels of representative nuclear encoded subunits of complexes V and II in skeletal muscle were higher in the TK2-deficient patient than in a control patient (153% and 401%, respectively for ATP5a and SDHB). By contrast, mitochondrial-encoded subunits (more prone to be affected by TK2 deficiency) from complexes III and IV and a nuclear encoded complex I subunit were less abundant in the TK2-deficient patient (78%, 56% and 58% for UQCRC2, COXII, and NDUFB8, respectively). Normal (control) levels of subunits representative of complexes I and IV were detected in the heart of the TK2-deficient patient (97% and 102%, respectively), whereas SDHB and UQCRC2 levels were lower (38 and 28%, respectively) and the level of complex V (ATP5α) was higher (126%). The greatest differences in the expression of OXPHOS subunits were detected in diaphragm tissue. Expression of several OXPHOS subunits was markedly lower in the TK2-deficient patient than in a control patient (33% for ATP5a and UQCRC2, 5% for SDHB and COXII and 3% for NDUFB8). Notably, Coomassie blue staining of the PVDF membrane demonstrated that while the overall protein load was similar between the patient and the control tissue (108%), the former produced a different protein profile, with changes in the major bands and an apparent loss of some proteins (Figure 2A).

UQCRC2 expression was undetectable in brain tissue of the TK2-deficient patient, and the abundance of the remaining OXPHOS subunits was lower than that of the control (5% for SDHB and COXII and 25% for NDUFB8) with the exception of ATP5α, which was slightly higher than control levels (116%). Analysis of liver samples revealed lower levels of ATP5a, UQCRC2, COXII and NDUFB8 (78%, 9%, 66% and 55%, respectively) in the TK2-deficient patient and normal SDHB (109%) levels (Figure 2B). Likewise, kidney levels of ATP5a, UQCRC2 and COXII were lower in the TK2-deficient patient than in the control (76%, 7% and 88% respectively), whereas SDHB and NDUFB8 content was higher (232% and 260%, respectively). Finally, near normal values of ATP5a, SDHB and NDUFB8 were detected in the small intestine of the TK2-patient (118%, 109% and 93%, respectively) whereas lower levels of the two mtDNA-encoded proteins, UQCRC2 and COXII, were observed (12 and 23%, respectively). These results demonstrated higher affectation of mitochondrial-encoded than nuclear OXPHOS subunits in these tissues, and higher SDHB suggested mitochondrial proliferation in kidney.

### 2.3. General Protein Profile in TK2 Deficiency

To gain insight into the different protein profiles observed in the diaphragm tissues, we first examined several proteins that should be constitutively expressed in all the tissues analyzed (Figure 3). We first assessed GAPDH, a key enzyme of glycolysis that is commonly used as a loading control for striated muscles. Western blotting showed that GAPDH expression in skeletal muscle, diaphragm and brain tissue was lower in the TK2-deficient patient than in the control (39%, 28% and 75% of control, respectively) whereas heart tissue showed normal levels (115%). Similarly, the levels of α-tubulin, a protein constituent of microtubules, were different between the patient’s tissues and the control tissues, with higher levels in patient skeletal muscle, heart, diaphragm and brain (110%, 301%, 147% and 133% of control, respectively). As anticipated, β-actin was not expressed in muscle, heart and control diaphragm, but strikingly it was detected in the patient’s diaphragm tissue, at 68% of the level found in the control brain sample (Figure 3). To confirm the presence of muscular cells in the patient’s diaphragm tissue, we tested for the myofibrillar protein myosin heavy chain β/slow, which is expressed by heart and slow fibers [17]. We found that the levels of myosin heavy chain β/slow were similar between the patient and the control (100 vs. 101%). By contrast, the abundance of myosin heavy chain β/slow was lower in the skeletal muscle (56% of control) and higher in the heart (289%) of the patient than in the control. Finally, as expected, it was not detected in any of the brain samples.

We next studied the different protein profiles observed in the patient and the control diaphragm tissue (see Figure 2A). After electrophoresis and Coomassie staining, we manually excised one stained band at 40 kDa that appeared to be absent in the patient sample but present in the control sample, and one major band at 65 kDa that was over-represented in the patient sample (Appendix A). Following digestion, the samples were subjected to MALDI-TOF/TOF mass spectrometry. Results of this analysis showed that the band at 40 kDa contained fructose-bisphosphate aldolase A, a key enzyme of glycolysis (Mascot top score 74) and the major band at 65 kDa contained albumin (Mascot top score 172).

To assess whether the dearth of proteins related to energy production in the patient’s diaphragm tissue was due to mechanical ventilation, we compared the levels of representative OXPHOS subunits and GAPDH in the patient’s diaphragm and several control diaphragms, including tissue from patients under ventilation at the time of death (marked as Cv in Figure 4A). Despite the high variability, we could clearly detect markers of all mitochondrial complexes, as well as GAPDH, in all control samples irrespective of the ventilated status of the individual. As expected, OXPHOS proteins were almost undetectable in the patient’s diaphragm tissue (Figure 4A,B). Also, the Coomassie-stained protein profile was similar between all controls and was different from the patient’s tissue.

We also confirmed the absence of β-actin and the presence of sarcomeric α-actin in the diaphragm tissue of ventilated controls but not in the patient’s sample, which showed the opposite expression pattern (Figure 5A,B). Finally, to discard the possibility that mechanical ventilation could downregulate TK2 expression and induce mitochondrial dysfunction in the diaphragm, we measured TK2 protein levels in all samples. Results showed variability in TK2 expression between the non-ventilated controls, but TK2 expression was clearly not affected by ventilation (167% vs. 100% for ventilated vs. non-ventilated controls). Reassuringly, TK2 protein was almost absent in the patient’s diaphragm sample (Figure 5A,B).

### 2.4. Oxidative Stress Markers in TK2 Deficiency

Oxidative stress is a common consequence of mitochondrial dysfunction, and protein oxidation has been reported to be especially high in those patients with most severe forms of myopathy [18]. To determine whether TK2 deficiency induces oxidative stress, we analyzed the levels of several antioxidant enzymes in tissues (Figure 6). Catalase expression was lower in the majority of the patient’s tissues than in control tissues, with the greatest difference observed in diaphragm tissue and no change in skeletal muscle. Peroxiredoxin 6 (PRDX6) levels were lower in diaphragm and liver tissue from the patient than from the control, but were more abundant in heart, kidney and small intestine. The cytosolic form of superoxide dismutase (cSOD) was highly overexpressed in skeletal muscle of the patient and was slightly higher in the heart and kidney, whereas diaphragm, brain and liver tissue had a lower content of this enzyme. No cSOD expression was detected in intestinal tissue from the control or the patient.

### 2.5. Protein Degradation Markers in TK2 Deficiency

Given the different protein profiles observed in diaphragm tissue between the TK2-deficient patient and controls, and that oxidative stress has been related to muscle atrophy and protein degradation [19], we investigated whether protein degradation was enhanced in the former by examining the levels of proteins involved in maintaining cellular proteostasis (ubiquitin-proteosome system and autophagy). We first analyzed the levels of p62, a multifunctional protein involved in the proteosomal degradation of ubiquitinated proteins. Western blotting showed that the abundance of p62 was greater in skeletal muscle, heart, and, particularly, in diaphragm, from the patient as compared with equivalent control tissues (213, 243, and 1496%, respectively), whereas its levels were low in brain (61%) (Figure 7). By contrast the autophagosome marker LC3II was almost undetectable in muscle, heart and diaphragm tissue of the patient and a control, but it was evident in brain, with similar values in the control and the patient tissue (Figure 7). Analysis of ubiquitinated proteins (Ubi-prot) revealed a lower abundance in the patient tissues with high p62 expression; specifically, muscle and diaphragm (32% and 74%, respectively). Furthermore, Ubi-prot was normal in the heart, but was higher in brain tissue from the patient than from the control (114% and 142%, respectively). Finally, we quantified the abundance of the catalytic subunits of the proteasome (α and β), finding that they were much more abundant in the patient’s diaphragm than in equivalent control tissue (2500% of the control value) and were slightly higher in the patient’s skeletal muscle in comparison with control tissue (225%).

To assess whether other protein degradation pathways were active in the patient’s diaphragm, we evaluated pro-caspase 3 and its active form. Whereas significant expression of pro-caspase 3 was detected only in the diaphragm tissue of ventilated controls and the patient (Appendix A), we failed to detect the active form in any sample. We also investigated the levels of the cysteine proteinase calpain 3 using two different monoclonal antibodies to assess the presence of autolysis fragments indicating activation of this protease (Appendix A). Using antibody 2C4, we observed full-length calpain 3 at 94 kDa in the patient’s diaphragm tissue, and two additional bands migrating at >100 kDa (also present in a non-ventilated control) and an additional band below 50 kDa that was also faintly detected in some control samples. With the second antibody, 12A2, we could detect only an autolysis product migrating at 55 kDa [20] in all the controls but not in the patient’s diaphragm tissue.

### 2.6. TK1 Levels in TK2 Deficiency

Finally, we sought to assess the levels of TK1, one of the enzymes responsible for the cytosolic nucleotide synthesis salvage pathway (Figure 8). Western blotting revealed TK1 expression in all of the tissue samples analyzed, migrating as oligomeric forms >100 and 140 kDa. Notably, the levels of TK1 were much more abundant in all the tissues from the patient than in equivalent control tissues (except brain), particularly in the patient’s diaphragm (Figure 8). Additional lower molecular weight forms were evident in the patient’s muscle, heart, diaphragm, kidney and intestine.

## 3. Discussion

Our goal was to provide the first analysis of the impact of TK2 deficiency on proteins of the OXPHOS system in adult *post mortem* tissues. Our results demonstrate that the p.Thr108Met mutation in TK2 induces differential changes to the protein component of the mitochondrial OXPHOS system in all studied tissues, with the most profound changes observed in the diaphragm, leading to the almost absence of proteins involved in energy-production systems—glycolysis and OXPHOS—as well as deficiency in some contractile proteins and abnormal expression of non-muscular proteins.

Respiratory failure is the main cause of death in patients of all ages with TK2 deficiency [3,4,5,9,10]. Diaphragmatic weakness occurs early in the course of the disease, usually before the loss of ambulation, and is the cause of the first medical consultation in many patients [8,21]. Our present results indicate that the diaphragm was much more affected by TK2 deficiency than skeletal muscle, as revealed by the greater loss in mtDNA content and in all OXPHOS proteins analyzed. Skeletal muscle also had a lower abundance of respiratory chain complexes I, III and IV than control tissue, but the finding of high levels of SDHB suggest the activation of compensatory mitochondrial proliferation in this tissue, which correlates with the muscle biopsy histological findings and the less pronounced mtDNA depletion compared with diaphragm. Multiple mtDNA deletions were also evident in skeletal muscle and also slightly detected in diaphragm, but were not observed in the remaining analyzed tissues. Heart seemed to be the least affected muscle in the patient, with lower complex II and III content (despite being rich in mitochondria) but normal mtDNA levels. Among the remaining tissues studied, brain exhibited low levels of all respiratory chain complexes tested despite normal mtDNA content. Liver was similar to skeletal muscle in terms of OXPHOS proteins and mtDNA depletion, but with no evidence of compensatory increase in mitochondria. Finally, kidney and gut showed alterations only in complex III and IV subunits, which were consistent with the lower mtDNA content in these tissues. Despite the evident OXPHOS protein loss or/and mtDNA depletion observed in brain, liver and kidney and, particularly, in intestine, there was nothing in the patient’s history or autopsy to suggest functional alterations in these systems and further studies are warranted. Perhaps the molecular changes were below a critical threshold to produce functional disturbances. The differences in mtDNA depletion and OXPHOS proteins observed between tissues could be partially attributed to the recently reported low levels of cytosolic and mitochondrial TK2 protein and activity in skeletal muscle, in comparison with other tissues observed in rats, which makes muscle the most dependent tissue on the salvage and *de novo* dTTP synthesis pathways [22]. Further research analyzing the dependence of diaphragm on these routes is still needed to assess if this muscle could be especially dependent on them. The present study correlates with previous findings in pediatric cases of TK2 deficiency describing low levels of mtDNA in autopsy samples of skeletal muscle [9,10]. The absence of histological alterations and mtDNA content in the patient’s brain also agree with some pediatric cases, although low mtDNA content associated to neuron loss, gliosis and ischemic injury have been described in other pediatric autopsy samples [9].

Studies in animal models have demonstrated that controlled invasive mechanical ventilation alters mitochondrial function. For instance, piglets mechanically ventilated for 5 days have reduced complex IV activity [23], and studies performed in rodents ventilated for 6–12 h have reported that diaphragmatic mitochondria have higher oxygen consumption in state 4 (non-ADP stimulated) than from control animals, lower respiratory control ratio, enhanced mitochondrial reactive oxygen species (ROS) production, disturbed dynamics and decreased activity of respiratory chain complexes [24,25,26,27,28,29,30], and these changes were associated with a decrease in the cross sectional area of the muscle fibers and lower specific force [25,26,29]. Studies analyzing the effects of mechanical ventilation on mitochondrial function in humans are scarce. Picard et al. [26] reported reduced complex II and IV and citrate synthase activities, reduced mitochondrial biogenesis, altered dynamics, and a higher frequency of mtDNA deletions in the diaphragm of brain-dead organ donors ventilated for 15 h to 1 week. A similar study demonstrated decreased expression of both nuclear and mitochondrial DNA-encoded complex IV subunits, but no alterations in mitochondrial content [31].

It is important to consider that the invasive mechanical ventilation used the aforementioned studies leaves the respiratory muscles‒the diaphragm‒totally inactive, whereas the NIV system allows a certain degree of diaphragm activity even when high pressures are applied [32]. In this regard, mechanical ventilation in critically ill patients, who have some diaphragm activity, has been reported to cause diaphragmatic atrophy in the absence of OXPHOS dysfunction [33]. Moreover, Hudson and coworkers [28] have reported that partial support ventilation in rats prevents mitochondrial-derived oxidative stress and preserves diaphragm protein synthesis during prolonged ventilation. Both of the ventilated control patients used in the present study, and specifically the patient on long-term NIV support, had lower expression of OXPHOS proteins than non-ventilated controls but OXPHOS levels were higher than in the TK2-deficient patient. Therefore, it seems that while the prolonged use of NIV may have had a mild impact on diaphragmatic mitochondrial function, the main inducer of the observed diaphragm OXPHOS deficiency is the TK2 mutation.

In addition to the loss of OXPHOS subunits in the patient’s diaphragm tissue, we also detected the loss of two key glycolytic enzymes, aldolase A and GAPDH, which suggests that glycolysis is also impaired in the diaphragm, essentially leaving this muscle with no energy production systems. We also observed an abnormal pattern of protein expression in this tissue, with a higher content of albumin and the unexpected high expression of non-muscle cytoskeletal β-actin [34]. Of note, previous studies in mouse models of muscular atrophy and Duchenne muscular dystrophy have demonstrated high levels of serum albumin in skeletal muscle and diaphragm that correlated with muscle degeneration [35,36,37]. This disturbance has been attributed to a combination of increased leakage of the protein from serum to the muscular interstitial space due to changes in vascular permeability, and a lower protein removal by lymphatic drainage [38], but not to endogenous albumin expression in muscle tissue [35]. This evidence suggests that in the present study albumin deposition on the patient’s diaphragm could occur by similar mechanisms. On the other hand, the expression of β-actin likely reflects compensatory reprogramming to re-express isoforms typical of immature muscle, myoblasts, to counteract the loss of the myofibrillar α-actin [34].

Mechanical ventilation in animal models and in humans is associated with diaphragmatic oxidative stress, which ultimately triggers tissue atrophy and dysfunction [24,28,30,31,33,39,40,41,42]. Mechanical ventilation-induced oxidative stress is attributed to the energy oversupply relative to the low energy demand, which promotes the reduction of the mitochondrial respiratory chain redox centers, leading to high ROS production, protein synthesis blockade and enhanced protein degradation [24,25,26,42]. A common consequence of primary mitochondrial disorders is impaired OXPHOS complexes assembly that, in turn, induces high ROS production and oxidative stress [18,43,44]. Therefore, mechanical ventilation-induced oxidative stress could have exacerbated the primary oxidative stress due to the mitochondrial defect in the TK-deficient patient. Although we did not examine oxidative stress markers, it seems plausible that the lower abundance of antioxidant enzymes in the patient’s diaphragm could be due to oxidative stress. In this regard, strong oxidative stress has been reported to overwhelm the capacity of some antioxidant enzymes, such as peroxiredoxins, leading to their inactivation and degradation [45], and therefore, to a vicious cycle of enhanced oxidative stress. This hypothesis of a highly oxidative environment in the patient’s diaphragm is further supported by the high levels of TK1, which has been reported to be upregulated by oxidative stress [46].

Several protein degradation pathways are induced by oxidative stress in ventilated diaphragm in both animal models and humans, including autophagy, the ubiquitin proteasome system (UPS), caspase 3, and the calcium-dependent activation of calpain 3 [25,28,30,31,39,40,42,47]. Ubiquitination is a cellular system that targets proteins for degradation by the autophagosome-lysosome system or by the UPS, or to be processed by non-proteolytic routes [48]. p62 is an adaptor for ubiquitin-tagged proteins targeted to the autophagosome-lysosome, but it may also bind the proteasome complex and is implicated in linking autophagy and the UPS [48]. In the present study, although the autophagosomal marker LC3II was only observed in brain, and was barely detectable in other tissues, the higher levels of p62 in the patient’s muscle and diaphragm correlated well with more abundant catalytic subunits of the proteasome and the lower content of ubiquitinated proteins in these tissues, supporting a coordinated activation of both pathways in the patient’s muscles leading to the observed protein loss. Interestingly, there is evidence of muscular dystrophy in skeletal muscle of patients harboring TK2 mutations [12], which could be related to the activation of protein degradation. Of note, it has been reported that proteins residing in the inner mitochondrial membrane, including OXPHOS subunits, can be degraded by the UPS [49,50,51]. We hypothesize that the strong upregulation of the UPS might play a role in the massive loss of OXPHOS subunits in the patient’s diaphragm. Regarding protein degradation dependent on specific proteases, we detected pro-caspase 3 and full-length calpain 3 in the patient’s diaphragm, but not their active forms. Although these results would indicate that the proteases were inactive in the patient’s diaphragm at the time of death, the loss of α-actin and aldolase A, which are substrates of caspase 3 and calpain 3, respectively [20,52], would suggest that active proteolysis by these enzymes may have occurred in previous stages of the disease. Of note, it has been described that aldolase A is associated to the triads in the muscle fibers where provides ATP for calcium uptake by the sarcoplasmic reticulum during relaxation, and regulates ryanodine receptor activity [53,54,55]. This evidence suggests that TK2 deficiency-induced aldolase A loss might lead to disturbances in the excitation-contraction-relaxation cycle in the diaphragm.

Preclinical studies have recently demonstrated that deoxypyrimidine nucleosides extend the lifespan of TK2-deficient mice (in both knock-in and knock-out models) by rescuing mtDNA depletion in skeletal muscle and other target tissues likely by boosting the cytosolic pyrimidine salvage pathway enzymes TK1 and dCK [13,14]. The efficacy of oral deoxynucleoside treatment is also supported by preliminary clinical studies, especially in patients with early-onset TK2 deficiency [5,21]. However, age-related metabolic changes in mice seem to influence the efficacy of this treatment, particularly by downregulation of TK1 activity. To the best of our knowledge, the evaluation of TK1 levels in adult human muscle has only been performed in one study [13], with the authors showing only a slight downregulation of the protein when compared with the expression in infant muscle, which would support the high probability of responding to treatment regardless of age [13]. In the present work, we observed that TK1 was overexpressed in all tissues of the patient, particularly the diaphragm, providing evidence at the cellular level that the TK1 salvage pathway is overactivated in the context of TK2 deficiency, and supporting the likelihood of response to nucleosides treatment also in adult patients.

We are aware that the present study has several experimental limitations. We did not directly measure oxidative stress markers, or analyze atrophy or protein synthesis pathways due to the limited availability of tissue. These processes may also be deregulated and play a role in the diaphragmatic dysfunction. In addition, we used a limited number of controls. As strength this is the first study to our knowledge on adult *post mortem* tissues of a TK2-deficient patient, and our findings provide insight into the pathophysiological bases of the disease and the applicability of new treatments currently under clinical investigation.

## 4. Materials and Methods

### 4.1. Tissue Collection

A clinical necropsy was requested with the authorization of the patient’s family, and *post mortem* tissues (quadriceps, skeletal muscle, heart, diaphragm, brain, liver, kidney‒renal cortex‒and gut) were obtained 1.5 h after death, immediately frozen in liquid nitrogen, and then stored at −80 °C until analysis.

*Post mortem* diaphragms from two patients with prolonged home NIV, and two nonventilated subjects who died from non-mitochondrial diseases were obtained as controls. One of the ventilated controls was a patient with SOD1-associated amyotrophic lateral sclerosis and a history of NIV overnight for 14 days, followed by 29 days of NIV 24 h/day due to worsening of the condition (Cv_1_, female, 37 years, and tissue obtained 1.5 h after death). The other ventilated control was a patient who underwent lung transplantation for severe COPD and who had been using NIV with high inspiratory pressures for 9 months, followed by 24 h with invasive mechanical ventilation (Cv_2_, female, age 64 years, diaphragm obtained 3 h after death). Both died from respiratory failure. Additionally, diaphragm samples from three patients who died of non-mitochondrial diseases without ventilation were obtained (one female and two male, 84, 83 and 88 years, diaphragms obtained 1.5, 2 and 3 h after death, C_1_, C_2_ and C_3_ respectively). Finally, skeletal muscles (quadriceps and psoas) from two controls were obtained (one male, one female; age 56 and 64 years; muscles obtained 6 and 3 h after death, respectively), as well as quadriceps, heart, brain, liver, kidney and gut from one of the non-ventilated non-mitochondrial patients (C_1_).

Control muscles and diaphragms and the associated data included in this study were provided by the ‘Biobanco i+12′ of the ‘Hospital 12 de Octubre’, which is integrated in the Spanish Hospital Biobanks Network (RetBioH; www.redbiobancos.es), following standard operation procedures with appropriate approval of the Ethical and Scientific Committees, reference number: 18/147 (28 May 2019) and 19/01(5 June 2019), respectively.

### 4.2. Tissue Processing

Tissue homogenates were prepared in ice-cold RIPA buffer (50 mM Tris-HCl pH 7.4, 1% NP-40, 0.5% Na-deoxycholate, 1% SDS, 150 mM NaCl, 2 mM EDTA) with protease and phosphatase inhibitors (Roche Diagnostics Corp., Indianapolis, IN, USA) using a Potter homogenizer with a tissue:buffer ratio 1:9 (weight:volume). Homogenates were maintained on ice for 30 min and centrifuged at 10,000× *g*, 30 min at 4 °C. Supernatants containing the solubilized proteins were collected and protein concentration was determined with the BCA Assay Kit (Pierce, Thermo Scientific, Waltham, MA, USA).

### 4.3. Western Blotting

Respiratory chain complex subunits, glycolytic and antioxidant enzymes, myofibrillar proteins, TK1, TK2 and proteins involved in protein degradation routes were analyzed by Western blotting. Samples of total homogenates containing 10–30 μg of protein were loaded onto SDS-PAGE gels (7.5–15%). For TK1 levels assessment, samples were treated with 0.2 mM dithiothreitol to disrupt large TK1 oligomers [56]. Resolved proteins were transferred to PVDF membranes, blocked with 5% skimmed milk or bovine serum albumin, incubated with the primary antibody (Appendix A) and then with a corresponding HRP-conjugated secondary antibody, and finally developed with the ECL Prime Western Blotting Detection Reagent (Amersham GE Healthcare, Little Chalfont, UK). Densitometry was performed with ImageJ (National Institutes of Health, Bethesda, MD, USA), and data were normalized by total protein load per well according to Coomassie blue staining of the membrane [57]. Images of the whole membranes showing molecular weight markers and each total protein assessment by Coomassie staining of the membranes are shown in Appendix A.

### 4.4. Proteomic Analysis

Twenty micrograms of total diaphragm tissue protein were incubated with 0.4 mM EDTA, 8%, lauryl dodecyl sulfate, 50 mM Tris–HCl pH 6.8, 4% glycerol, 0.075% Serva Blue G250 and 0.025% Phenol Red for 10 min at room temperature. Samples were then separated by electrophoresis on a 12% polyacrylamide gel, which was subsequently stained with Brilliant Blue G-Colloidal Concentrate (Merck, Sigma-Aldrich, Darmstadt, Germany) allowing visualization of separated proteins [58].

Bands were manually excised and digested in-gel with trypsin. Gel pieces were washed three times with 25 mM ammonium bicarbonate/50% acetonitrile and dried in a Speed-Vac. Then, 10 μL of 25 μg/mL modified bovine trypsin (ABSciex, Framingham, MA, USA) in 25 mM ammonium bicarbonate was added to the dried gel pieces and the samples were incubated overnight at 37 °C. Peptides were extracted three times with 10% formic acid/50% acetonitrile followed by lyophilization in a Speed-Vac. Peptides were then resuspended in an acetonitrile/formic acid solution and mixed (1:1) with a matrix consisting of α-cyano-4-hydroxycinnamic acid and aliquots were spotted onto MALDI sample target plates.

Peptide mass spectra were obtained on a MALDI-TOF/TOF mass spectrometer (Applied Biosystems 4800 Proteomics Analyzer, Applied Biosystems, Foster City, CA, USA) in the positive ion reflector mode. Spectra were obtained across the mass range 700–4500 Da with ~1500 laser shots. A data-dependent acquisition method was created to select the six most intense peaks from each sample, excluding those from the matrix, trypsin autolysis, or acrylamide peaks, for subsequent MS/MS data acquisition. Trypsin autolysis peaks were used for internal calibration of the mass spectra, allowing a routine mass accuracy >20 ppm. Spectra were processed and analyzed on the Global Protein Server Workstation (Applied Biosystems), which uses internal Mascot v2.1.1 software (Matrix Science Ltd., London, UK) for searching the peptide mass fingerprints and MS/MS data.

Searches were performed against SwissProt (release date 010111) under all taxonomic categories and the following parameters: (i) two missed cleavages by trypsin; (ii) mass tolerance of precursor ions 25 ppm and product ions 0.3 Da; and (iii) oxidation of methionine as variable modification. Mascot scores > 56 were considered significant (*p* < 0.05).

### 4.5. Mitochondrial DNA Studies

Tissue DNA was isolated with the DNA Mini Kit (Qiagen, Hilden, Germany). DNA was examined for the presence of multiple mtDNA deletions by long-range polymerase chain-reaction (LR-PCR) through amplification of the entire 16.5 kb mtDNA using TaKaRa LA Taq DNA Polymerase (Takara-Bio, Shiga, Japan) (primers and PCR conditions available on request). PCR products were electrophoresed in 0.8% agarose gels.

Mitochondrial DNA content was calculated as the relative quantification of mtDNA versus nuclear DNA (nDNA) copy-number using TaqMan probes against the MTRNR1 and RNAseP genes, respectively, on an Applied Biosystems 7500 Real-Time PCR System (Applied Biosystems, Foster City, CA, USA) [59].

## Figures and Tables

**Figure 1 ijms-22-05598-f001:**
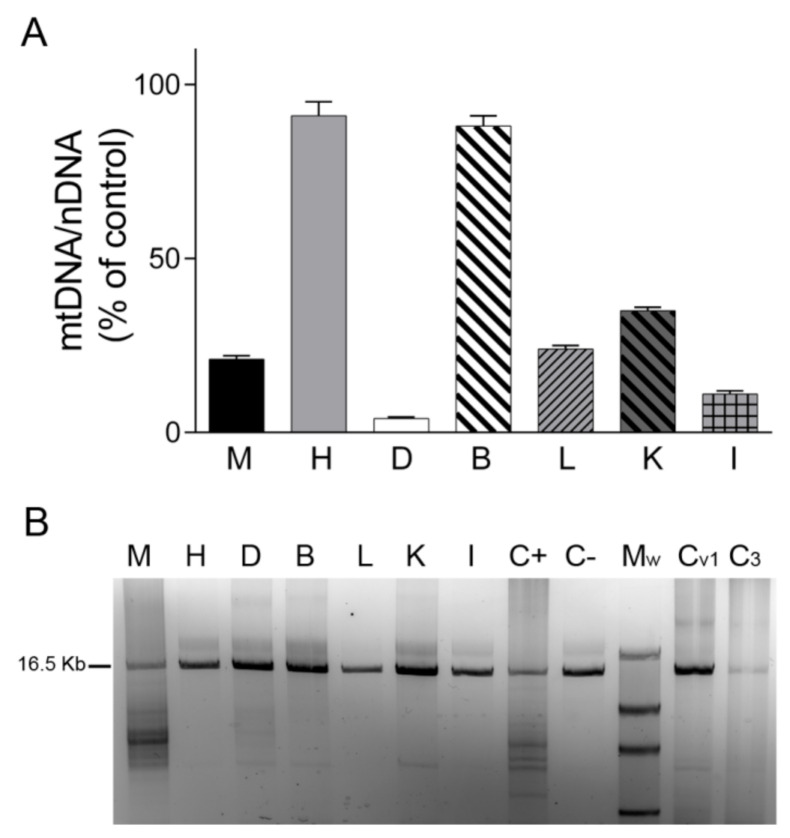
mtDNA content and mtDNA deletions. (**A**) mtDNA/nDNA copy-number ratio analyzed by real-time PCR in different tissues of the patient expressed as a percentage of the corresponding control tissue. Data are mean ± SD of three independent determinations. (**B**) Agarose gel showing mtDNA amplified by long-range PCR in different tissues of the patient and in control diaphragms. Normal sized amplicon of 16.5 kb is indicated. Lower molecular weight bands are suggestive of mtDNA deletions. A lower-size band of presumably non-specific amplification can be observed in B, K and controls. Abbreviations: M, skeletal muscle; H, heart; D, diaphragm; B, brain; L, liver; K, kidney; I, small intestine; C+, skeletal muscle positive control for mtDNA deletions; C−, skeletal muscle negative control for mtDNA deletions; Mw, Roche DNA Molecular Weight Marker II (Merck, Sigma-Aldrich, Darmstadt, Germany); Cv_1_, ventilated control diaphragm 1; C_3_, non-ventilated control diaphragm 3.

**Figure 2 ijms-22-05598-f002:**
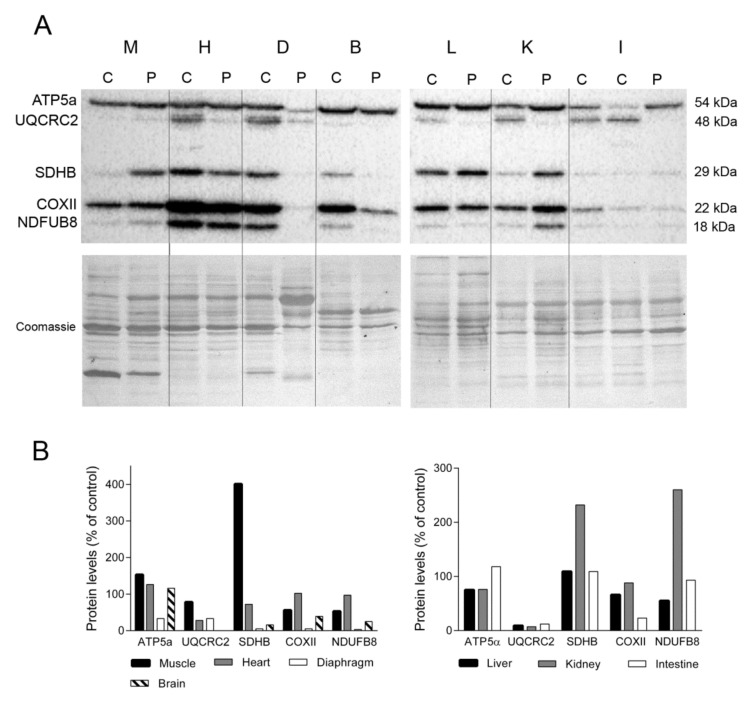
OXPHOS system subunits. (**A**) Representative Western blotting of OXPHOS system subunits in tissue homogenates of a control (C) (n = 3 for muscle, n = 1 for other tissues) and the patient (P). Shown below is a representative Coomassie blue-stained gel. (**B**) Densitometry analysis: protein levels were normalized to total protein loading according to Coomassie blue staining. Data are expressed as percentage of control values. Abbreviations: M, skeletal muscle; H, heart; D, diaphragm; B, brain; L, liver; K, kidney; I, intestine. Two samples of control intestine are shown; large (left) and small (right). ATP5a, ATP synthase subunit alpha; UQCRC2, ubiquinol-cytochrome C reductase core protein 2; SDHB, succinate dehydrogenase complex iron sulfur subunit B; COXII, cytochrome c oxidase subunit II; NDUFB8, NADH:ubiquinone oxidoreductase subunit B8.

**Figure 3 ijms-22-05598-f003:**
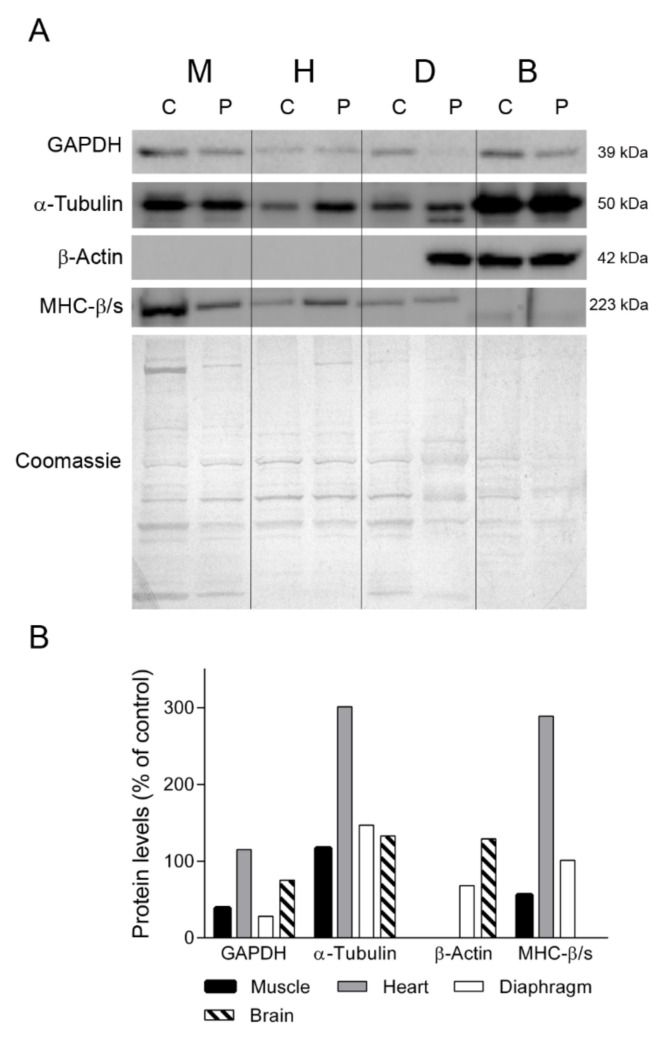
Constitutive proteins. (**A**) Representative western blotting of proteins with constitutive expression in tissue homogenates of a control (C) (n = 3 for muscle, n = 1 for other tissues) and the patient (P). Shown below is a representative Coomassie blue-stained gel. (**B**) Densitometry analysis: protein levels were normalized to total protein loading according to Coomassie blue staining. Data are expressed as percentage of control values. Abbreviations: M, skeletal muscle; H, heart; D, diaphragm; B, brain. GAPDH, glyceraldehyde-3-phosphate dehydrogenase; MHC-β/slow, myosin heavy chain beta/slow.

**Figure 4 ijms-22-05598-f004:**
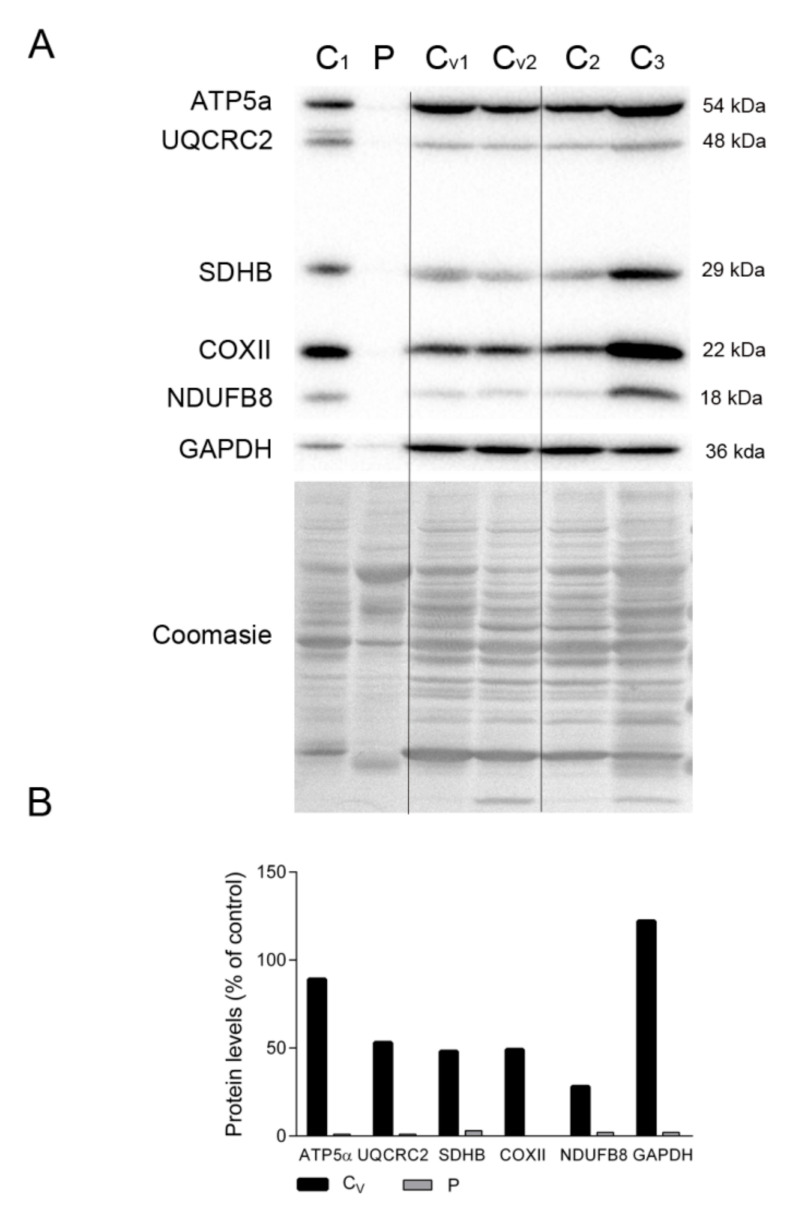
OXPHOS system subunits in patient and ventilated control diaphragms. (**A**) Representative Western blotting of OXPHOS system subunits in diaphragm homogenates of non-ventilated controls (C, n = 3), ventilated controls (Cv, n = 2), and the patient (P). Shown below is a representative Coomassie blue-stained gel. (**B**) Densitometry analysis: protein levels were normalized to total protein loading according to Coomassie blue staining. Data are expressed as percentage of non-ventilated control values. Abbreviations: ATP synthase subunit alpha; UQCRC2, ubiquinol-cytochrome C reductase core protein 2; SDHB, succinate dehydrogenase complex iron sulfur subunit B; COXII, cytochrome c oxidase subunit II; NDUFB8, NADH:ubiquinone oxidoreductase subunit B8; GAPDH, glyceraldehyde-3-phosphate dehydrogenase.

**Figure 5 ijms-22-05598-f005:**
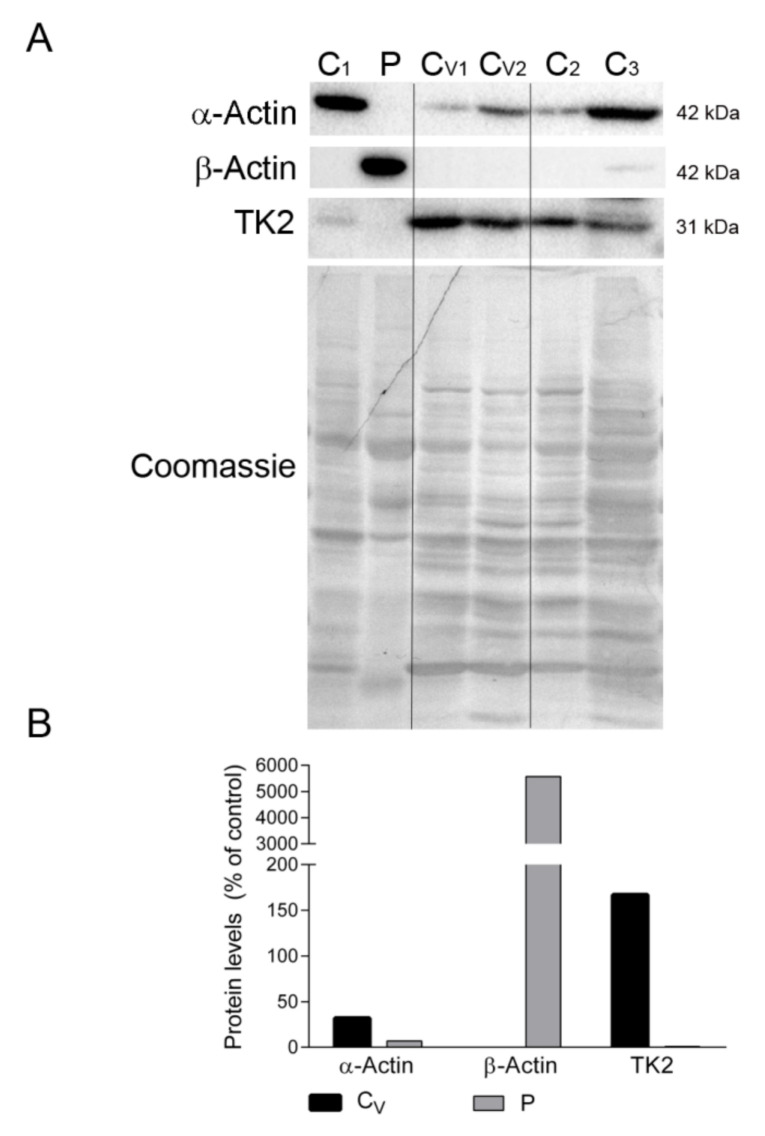
Actin isoforms and TK2 in diaphragm. (**A**) Representative Western blotting of skeletal muscle α-actin, microtubule β-actin and TK2 in diaphragm homogenates of non-ventilated controls (C) (n = 3), ventilated controls (Cv, n = 2), and the patient (P). Shown below is a representative Coomassie blue-stained gel. (**B**) Densitometry analysis: protein levels were normalized to total protein loading according to Coomassie blue staining. Data are expressed as percentage of non-ventilated control values. Abbreviations: TK2, thymidine kinase 2.

**Figure 6 ijms-22-05598-f006:**
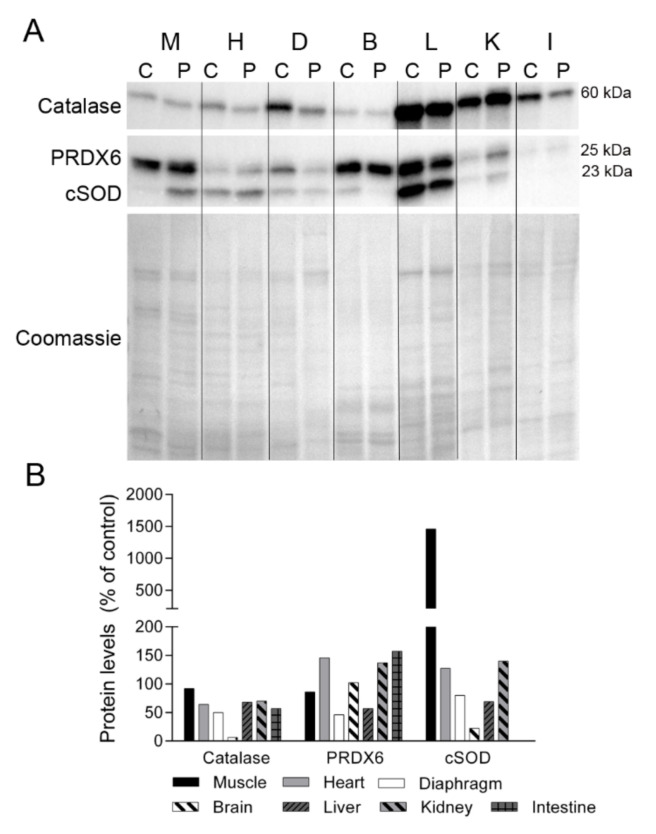
Antioxidant enzymes. (**A**) Representative Western blotting of antioxidant enzymes in tissue homogenates of a control (C) (n = 1) and the patient (P). Shown below is a representative Coomassie blue-stained gel. (**B**) Densitometry analysis: protein levels were normalized to total protein loading according to Coomassie blue staining. Data are expressed as percentage of control values. Abbreviations: M, skeletal muscle; H, heart; D, diaphragm; B, brain; L, liver; K, kidney; I, small intestine. PRDX6, peroxiredoxin 6; cSOD, cytosolic superoxide dismutase.

**Figure 7 ijms-22-05598-f007:**
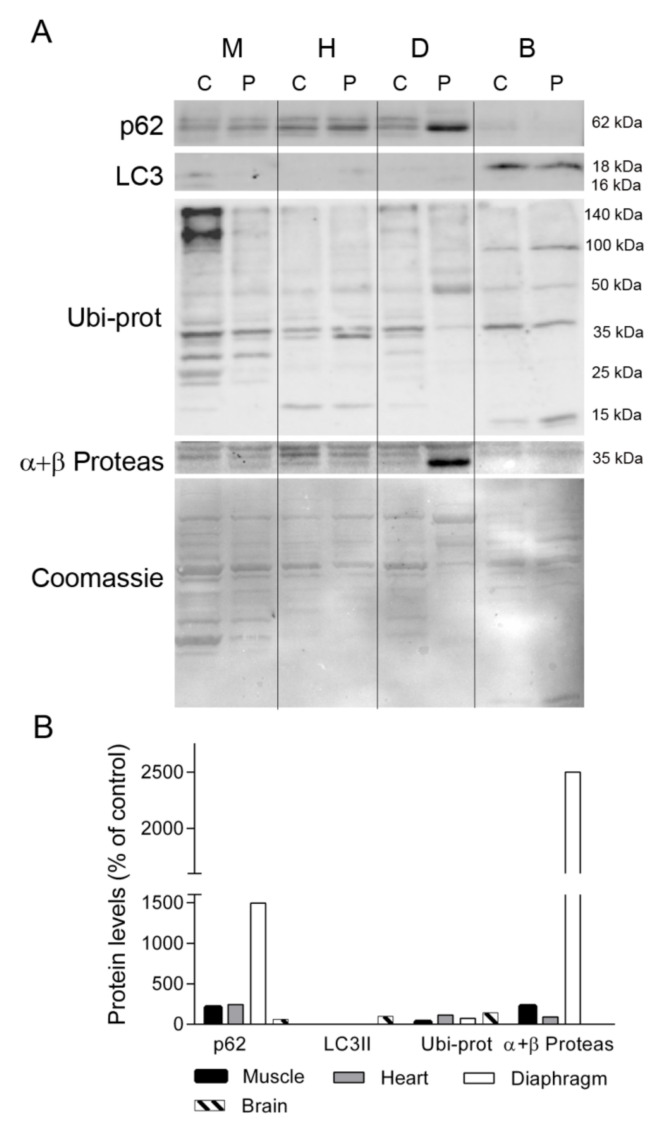
Autophagy and ubiquitin-proteasome system markers. (**A**) Representative Western blotting of autophagy and ubiquitin proteasome markers in tissue homogenates of a control (C, n = 1) and the patient (P). Shown below is a representative Coomassie blue-stained gel. (**B**) Densitometry analysis: protein levels were normalized to total protein loading according to Coomassie blue staining. Data are expressed as percentage of control values. Abbreviations: M, skeletal muscle; H, heart; D, diaphragm; B, brain; L, liver; K, kidney; I, small intestine. LC3, microtubule-associated proteins 1A/1B light chain 3A; Ubi-prots, ubiquitinated proteins; α + β Proteas, α and β catalytic subunits of the 26S proteasome.

**Figure 8 ijms-22-05598-f008:**
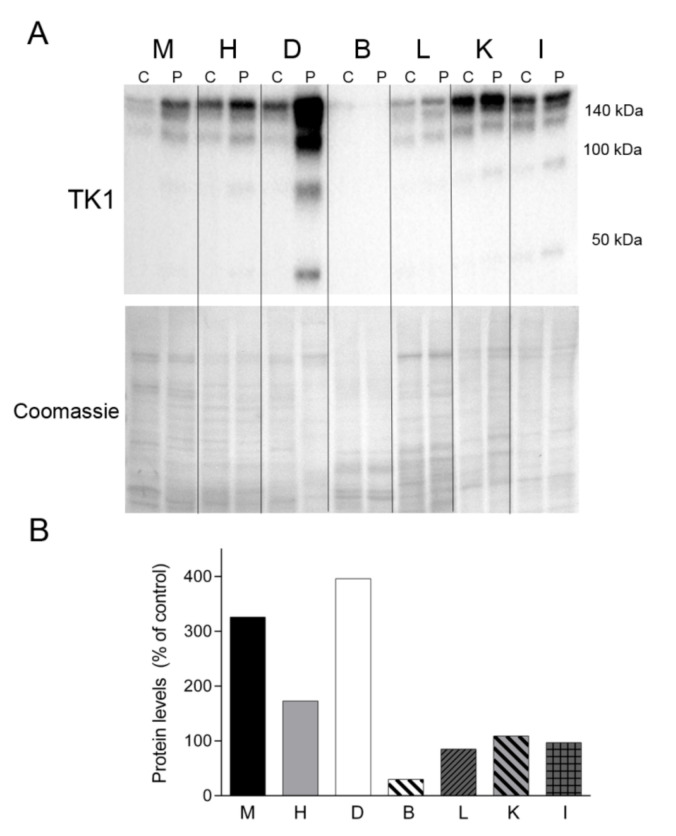
TK1 levels. (**A**) Western blotting of TK1 in tissue homogenate samples treated with dithiothreitol from a control (C) (n = 1) and the patient (P). Coomassie straining of the membranes is shown as total protein loading control. (**B**) Densitometry analysis: protein levels were normalized to total protein loading according to Coomassie blue staining. Data are expressed as percentage of control tissue Abbreviations: M, skeletal muscle; H, heart; D, diaphragm; B, brain; L, liver; K, kidney; I, small intestine. TK1, thymidine kinase 1.

## Data Availability

The data that support the findings of this study are available from the corresponding author upon reasonable request.

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
