# Peer review of "Preferent Diaphragmatic Involvement in TK2 Deficiency: An Autopsy Case Study"

_ijms, 2021, doi:10.3390/ijms22115598_

Round 1
Reviewer 1 Report
Authors presented an interesting manuscript related to post mortem multiple tissue analysis of patient with TK2 deficiency. Authors applied various approaches: molecular-genetic analysis of mtDNA maintenance (deletion, copy number) and protein investigation of mitochondrial status (western blotting, proteomic analysis). Paper is unique due to wide spectrum of tissue analysed as well as origin of tissue from patient with mitochondrial disease due to TK2 mutation. Authors found not previously published tissue specific impact of mutation, specifically to diaphragm, that is not standardly investigated specific type of muscle tissue. This information is very useful for all specialists that are involved in diagnostics of mitochondrial disorders. Data related to ventilation and the impacts on the diaphragm were also interesting.
I have a few comments:
Results
Page 2, line 74-77 „Analysis of mtDNA integrity using LR-PCR revealed mtDNA deletions only in the skeletal muscle and diaphragm of the TK2-deficient patient (in apparently low and high proportions, respectively) and not in the liver, kidney and small intestine. “
But in figure 1B, I can detect deleted bands in B and K…? As well as slight bands in C3…
Generally, all WB in manuscript are normalised to Coomassie blue total loading protein, but it is common to give as a control any of the proteins detected by WB on the same membrane.
It will be useful to mark KDa of analysed proteins on the right side of each WB figure.
Page 4, lines 123-124
„These results demonstrated higher affectation of mi1tochondrial-encoded than nuclear OXPHOS subunits in these tissues, and higher SDHB suggested mitochondrial proliferation in kidney. “
What part of the kidney is it ..?
Page 6, lines169-170
„Results of this analysis showed that the band at 40 kDa contained fructose-bisphosphate aldolase, a key enzyme of glycolysis 170 (Mascot top score 74) and the major band at 65 kDa contained albumin (Mascot top score 172) “
Can the authors speculate what are the probable reasons for the high levels of albumin in the diaphragm?
Figure 7
LC3 in figure versus LC3B in image caption.
Are all displayed WBs from one identical gel? Or from multiple gels? Then it would require the control protein to be given for each gel / WB.
Discussion
Lines 360-363 „.. it seems plausible that the lower abundance of antioxidant enzymes in the patient’s diaphragm could be due to oxidative stress, as long term oxidative stress leads to depletion of antioxidants [35]. Of note, the higher levels of TK1 in all the studied tissues of the patient, but particularly in the diaphragm, support the presence of oxidative stress, which induces the enzyme [36].“
Can the authors describe more in details what they mean...?
Discussion is well readable, but a bit vague and descriptive, it would like to point out some interesting results - for example, the authors could speculate more about the finding : ..“loss of the enzyme fructose-bisphosphate aldolase, and enrichment for serum albumin in the patient’s diaphragm tissue.“
Materials and methods:
The time between the hour of death and tissue collection is missed...
Lines 458-462 „Additionally, skeletal muscles (psoas and quadriceps) from 3 controls were obtained, as well as heart, brain, liver, kidney and gut from a non-ventilated non-mitochondrial patient. Control muscles and diaphragms were obtained from the tissue Biobank of the Hospital Universitario 12 de Octubre after ethical approval….“
Age and gender of these „control“ tissues are not given.
Author Response
In response to the specific points of Reviewer 1.
We would like to thank the Reviewer for his/her thoughtful and constructive comments. We have considered all of the reviewers’ recommendations, and have incorporated them into the revised manuscript. Please, see below our responses to the raised queries and comments.
Results
Q1. Page 2, line 74-77. Analysis of mtDNA integrity using LR-PCR revealed mtDNA deletions only in the skeletal muscle and diaphragm of the TK2-deficient patient (in apparently low and high proportions, respectively) and not in the liver, kidney and small intestine”. But in figure 1B, I can detect deleted bands in B and K…? As well as slight bands in C3…
R1. We considered multiple mtDNA deletions when several electrophoretic bands or a smeared fuzzy band lower than wild-type mtDNA size was observed in a particular tissue. In those tissues in which we only detected a single-band of low molecular size, as occurred in brain, kidney, and controls, we did not attribute that band to a mtDNA deletions, it was considered a potential artifact due non-specific PCR amplification since this band runs at a similar molecular weight in B and K, as well as in controls. We have added a clarification on this band in the Figure 1 legend.
Q2. Generally, all WB in manuscript are normalised to Coomassie blue total loading protein, but it is common to give as a control any of the proteins detected by WB on the same membrane. It will be useful to mark KDa of analysed proteins on the right side of each WB figure.
R2. According to reviewer’s comment, we have marked the kDa of each analyzed protein on each WB figure. We agree with the reviewer that a loading control by immunodetection of a specific protein with constant expression among the studied samples can be a useful approach to compare the total protein loading and transfer efficiency in western blotting procedures. However, in the field of mitochondrial disorders is very common to find differences in the protein levels of the commonest loading controls, such as GAPDH, α and β actin and tubulin, because the expression of these proteins is secondarily affected by the mitochondrial defect. For example, GAPDH can change due to a compensatory mechanism of up-regulation of glycolysis enzymes to obtain ATP. Also, actins and tubulins levels can change due to sarcomere alterations and the close interaction of mitochondria with the cytoskeleton. Several proteomic studies performed in our laboratory analyzing the protein profile of skeletal and cardiac muscles of a mouse model of mitochondrial complex I deficiency, have revealed differences in the levels of these common loading controls between control and mitochondrial deficient mice (unpublished data, draft manuscript in writing). In fact, in the present study the first clue to the highly altered protein profile observed in the patient’s diaphragm was the low level of GAPDH observed that leaded as to analyze other common loading controls, (included in the manuscript as Figure 3 constitutive proteins) and to the proteomic analysis of this tissue. For these reasons, we performed an alternative loading control using the whole membrane staining with the Coomassie blue technique reported by Welinder and Ekblad, 2001 (mentioned in the materials and methods section), that solves all the limitations of detecting only a protein as loading control. This approach has become a routine now in our laboratory and we have published a study using this technique [Fiuza-Luces et al., 2019 doi: 10.3389/fneur.2019.00790]. Moreover, some companies such as BioRad, offers pre-cast gels that allows total protein staining to be used as loading control instead of single housekeeping protein detection https://www.bio-rad.com/es-es/applications-technologies/stain-free-imaging-technology?ID=NZ0G1815
Q3. Page 4, lines 123-124. These results demonstrated higher affectation of mi1tochondrial-encoded than nuclear OXPHOS subunits in these tissues, and higher SDHB suggested mitochondrial proliferation in kidney. “What part of the kidney is it ...?
R3. The part of the kidney analyzed in the study was renal cortex; this detail has been added in the materials and methods section.
Q4. Page 6, lines169-170.Results of this analysis showed that the band at 40 kDa contained fructose-bisphosphate aldolase, a key enzyme of glycolysis 170 (Mascot top score 74) and the major band at 65 kDa contained albumin (Mascot top score 172) “. Can the authors speculate what are the probable reasons for the high levels of albumin in the diaphragm?
R4. Following the reviewer’s request in the discussion we have included a comment on the putative reason of the high albumin levels observed in the diaphragm.
Q5. Figure 7. LC3 in figure versus LC3B in image caption.
R5. We have corrected the caption.
Q6. Are all displayed WBs from one identical gel? Or from multiple gels? Then it would require the control protein to be given for each gel / WB.
R6. Several proteins with different molecular weights were immunodetected in the same membrane when the acrylamide percentage of the gel allowed appropriate separation of the protein bands. In other figures we had to use different gels, and membranes to perform optimal protein detection. In this latter case, we did not add all the images showing the Coomassie staining for each membrane since they take up a large amount of space in the figures and the loading was similar between samples in all of the membranes. According to the reviewer suggestion in the revised version of the manuscript we have provided all the whole images showing the Coomassie staining in a supplementary figure: Figure S4.
Discussion
Q7. Lines 360-363. it seems plausible that the lower abundance of antioxidant enzymes in the patient’s diaphragm could be due to oxidative stress, as long-term oxidative stress leads to depletion of antioxidants [35]. Of note, the higher levels of TK1 in all the studied tissues of the patient, but particularly in the diaphragm, support the presence of oxidative stress, which induces the enzyme [36]. “
Can the authors describe more in details what they mean...?
R7. Following reviewer’s query, we have clarified our statement.
Q8. Discussion is well readable, but a bit vague and descriptive, it would like to point out some interesting results - for example, the authors could speculate more about the finding: “…loss of the enzyme fructose-bisphosphate aldolase, and enrichment for serum albumin in the patient’s diaphragm tissue.“
R8. We have added a speculative proposal for the fructose-bisphosphate aldolase loss observed in the patient’s diaphragm as well a comment on previous studies demonstrating serum albumin infiltration in damaged muscles and diaphragm.
Materials and methods:
Q9. The time between the hour of death and tissue collection is missed...
R9. The time between death and necropsy for each control and the patient has been added in the material and methods section.
Q10. Lines 458-462. Additionally, skeletal muscles (psoas and quadriceps) from 3 controls were obtained, as well as heart, brain, liver, kidney and gut from a non-ventilated non-mitochondrial patient. Control muscles and diaphragms were obtained from the tissue Biobank of the Hospital Universitario 12 de Octubre after ethical approval….“ Age and gender of these „control“ tissues are not given.
R10. Following the reviewer’s request we have included this information in the revised manuscript.
Reviewer 2 Report
Laine-Menéndez et al., describe the autopsy findings of a patient with late-onset thymidine kinase 2 (TK2) deficiency who died of respiratory failure. The article is well written and provides relevant information about this pathology. Suggestions to improve the article are the following:
Introduction
Please include the OMIM number of the TK2 gene in the introduction.
The article by Mazurova S et al., Cardiol Young. 2017 Jul;27(5):936-944. and Götz A et al., Brain. 2008 Nov;131(Pt 11):2841-50. contain the partial description of the autopsy in several TK2 patients. Since the strong point of the article is the autopsy findings, these citations with their findings should be included in the introduction and the discussion section.
New phenotypes of the disease should be mentioned in the introduction (de Fuenmayor-Fernández de la Hoz CP et al., Mol Genet Metab Rep. 2021 Jan 6;26:100701)
Results
The clinical summary should appear in the first part of the Results section.
Subtitles must be provided in the results section to facilitate reading.
Discussion
Line 344 write energy production systems instead of energy producing systems
Line 408, include “adult” patient. There are several postmortem studies in TK2 children
Functional evidence of diagrammatic dysfunction in TK2 patients should be included in the discussion. (Barcia G et al. Neuromuscul Disord. 2020 Jul;30(7):593-598.)
The results of this study should be compared with those obtained in TK2 animal models. (Wang L et al., BMC Mol Cell Biol. 2020 Apr 28;21(1):33.)
Materials and Methods
Line 419 write Her sister
Line 435, Authors should include images of muscle biopsies, at least as supplementary material.
Author Response
In response to the specific points of Reviewer 2
We would like to thank the Reviewer for his/her thoughtful and constructive comments. We have considered all of the reviewers’ recommendations, and have incorporated them into the revised manuscript. Please, see below our responses to the queries raised.
Introduction
Q1. Please include the OMIM number of the TK2 gene in the introduction.
R1. According to the reviewer request the OMIM is included in the revised version of the manuscript.
Q2. The article by Mazurova S et al., Cardiol Young. 2017 Jul;27(5):936-944. and Götz A et al., Brain. 2008 Nov;131(Pt 11):2841-50. contain the partial description of the autopsy in several TK2 patients. Since the strong point of the article is the autopsy findings, these citations with their findings should be included in the introduction and the discussion section.
R2. We have added these two references in the introduction and discussion section.
Q3. New phenotypes of the disease should be mentioned in the introduction (de Fuenmayor-Fernández de la Hoz CP et al., Mol Genet Metab Rep. 2021 Jan 6;26:100701).
R3. This study has been mentioned in the introduction.
Results
Q4. The clinical summary should appear in the first part of the Results section.
R4. Done
Q5. Subtitles must be provided in the results section to facilitate reading.
R5. We have added 6 sections with subtittles in the results according to the reviewer’s suggestion with the clinical summary as first section
Discussion
Q5. Line 344 write energy production systems instead of energy producing systems
R5. Corrected
Q6. Line 408, include “adult” patient. There are several postmortem studies in TK2 children
R6. According to the reviewer’s query we have clarified in the manuscript that this is the first study in post mortem tissues of an adult with TK2 deficiency.
Q7. Functional evidence of diagrammatic dysfunction in TK2 patients should be included in the discussion. (Barcia G et al. Neuromuscul Disord. 2020 Jul;30(7):593-598.)
The results of this study should be compared with those obtained in TK2 animal models. (Wang L et al., BMC Mol Cell Biol. 2020 Apr 28;21(1):33.)
R7. Both articles have been included in the discussion.
Materials and Methods
Q8. Line 419 write Her sister
R8. Done. We have replaced: ‘.The sister had had difficulties …’ by ‘Her sister had had difficulties..?
Q9. Line 435, Authors should include images of muscle biopsies, at least as supplementary material.
R9. We have added a Supplementary Figure (Figure S1) showing muscle biopsies findings.
Round 2
Reviewer 1 Report
The authors sufficiently answered all questions and completed the manuscript. Accept the article in present form .